# Study of High-Density Polyethylene (HDPE) Kinetics Modification Treated by Dielectric Barrier Discharge (DBD) Plasma

**DOI:** 10.3390/polym12102422

**Published:** 2020-10-21

**Authors:** João Freire de Medeiros Neto, Ivan Alves de Souza, Michelle Cequeira Feitor, Talita Galvão Targino, Gutembergy Ferreira Diniz, Maxwell Santana Libório, Rômulo Ribeiro Magalhães Sousa, Thercio Henrique de Carvalho Costa

**Affiliations:** 1Plasma Materials Processing Laboratory, Federal University of Rio Grande do Norte, Natal 59078-970, Brazil; joaonetofm@ufrn.edu.br (J.F.d.M.N.); mcfeitor@gmail.com (M.C.F.); talitagalvao@ufrn.edu.br (T.G.T.); maxwellsantana@ect.ufrn.br (M.S.L.); 2Department of Mechanical Engineering, Federal University of Piauí, Teresina 64049-550, Brazil; ivanalves@ufrn.edu.br (I.A.d.S.); romulorms@gmail.com (R.R.M.S.); 3Department of Mechanical Engineering, Federal University of Uberlândia, Uberlândia 38408-100, Brazil; gutembergyferreira@gmail.com

**Keywords:** DBD plasma, HDPE, Langmuir, wettability

## Abstract

In this work, the plasma was used in the dielectric barrier discharge (DBD) technique for modifying the high-density polyethylene (HDPE) surface. The treatments were performed via argon or oxygen, for 10 min, at a frequency of 820 Hz, voltage of 20 kV, 2 mm distance between electrodes, and atmospheric pressure. The efficiency of the plasma was determined through the triple Langmuir probe to check if it had enough energy to promote chemical changes on the material surface. Physicochemical changes were diagnosed through surface characterization techniques such as contact angle, attenuated total reflection to Fourier transform infrared spectroscopy (ATR-FTIR), X-ray excited photoelectron spectroscopy (XPS), and atomic force microscopy (AFM). Plasma electronics temperature showed that it has enough energy to break or form chemical bonds on the material surface, impacting its wettability directly. The wettability test was performed before and after treatment through the sessile drop, using distilled water, glycerin, and dimethylformamide, to the profile of surface tensions by the Fowkes method, analyzing the contact angle variation. ATR-FTIR and XPS analyses showed that groups and bonds were altered or generated on the surface when compared with the untreated sample. The AFM showed a change in roughness, and this directly affected the increase of wettability.

## 1. Introduction

High-density polyethylene (HDPE) is one of the commonly used polymers in industrial, medical, and biomedical applications, as it presents: mechanical properties, flexibility, and high chemical stability [1,2,3,4,5]. However, a problem associated with HDPE is that it has a partially hydrophobic surface, due to its low surface energy and lack of functional groups, resulting in limited adhesion and low chemical reactivity [6,7,8].

In the last decades, there was an increase in the number of technologies to superficial modification of materials that sought to increase hardness, wear resistance, adhesion strength, hydrophilicity, biocompatibility, among other materials’ properties [9,10,11,12]. Among these technologies, the treatment of plasma in dielectric barrier discharge (DBD) has been used to increase the surface energy of polymeric materials by incorporating polar groups on the surface without changing its mass composition [13,14,15,16,17]. The process occurs between two electrodes, where at least one of which must be covered by a dielectric [18,19,20]. An advantage over other plasma formation techniques is that DBD does not require low pressures, avoiding the use of vacuum systems and the use of high electrical currents, reducing energy consumption, and making it a low-cost process [21,22].

The modifications through the DBD plasma can be chemical, promoting the dissociation of bonds such as C–C or C–H and thus allowing the formation of hydrophilic character groups, associated with oxygen species. Otherwise, physical modification is possible, through the etching process of species, promoting changes in surface roughness. Despite having different characteristics, chemical and physical changes directly affect the wettability and surface adhesion of the material. [21,23,24,25].

Several attempts have been reported in the literature using DBD plasma to improve the wettability of polymeric surfaces, since it does not deteriorate and can modify the treated surface [17,23,25,26,27]. The authors justify that the increase in wettability in polymeric materials is directly associated with the incorporation of functional groups such as C–O, C=O, and O–C–O [11,28,29,30,31]. The incorporated functional groups were verified, by the authors, using the XPS or FTIR techniques, with the associated reduction of the peak of the C–C and C–H bonds [5,12,32]. These modifications have been made in polymers such as: polypropylene [11,31], polyethylene [3,12,32,33], poly (lactic acid) [4,34], polyester [9,26,35], and others [10,29,36,37,38,39,40].

The use of polyethylene surface modification techniques results in increased wettability, from the interaction with functional groups and morphological modification [41]. Researches have obtained an increase in the wettability of polyethylene using techniques such as plasma jet, direct fluorination, UV-radiation, leisure, and others [23,42,43,44,45,46]. The use of techniques reported the reduction of C–C and C–H bindings from the inclusion of oxygen and nitrogen in the surface [3,4,5,23]. It is also reported that the introduction of an inert gas facilitates the formation of plasma and promotes the dissociation of hydrocarbons [47,48].

Therefore, it is observed that several studies have investigated the high-density polyethylene wettability increase. However, DBD plasma has been little explored when compared to the high potential of this technique. Thus, this study will investigate the effect of this treatment on the HDPE surface properties based on the assessment of physicochemical changes and surface wettability. So, it is possible to assess how the treatment effect is optimized for applications where greater material adherence is required, since the HDPE initially does not have functional groups, making this type of interaction difficult.

## 2. Materials and Methods

### 2.1. Materials

Square shaped samples with 15 mm sides and 1.5 mm thickness were prepared using high-density polyethylene (HDPE) disk, processed by injection molding, supplied by the GFG PLASTICS FABRICATIONS LTD, Hull, United Kingdom. In thi.s work, 5 samples were treated to each treatment condition and the results showed are the average obtained in the analysis.

### 2.2. Plasma Reactor

The used reactor (Figure 1) consisted of an acrylic tube, closed by two Teflon (PTFE) flanges, at the top and bottom of the tube. An anode electrode was placed on the top flange, and a cathode was placed on the bottom. A disk-shaped alumina dielectric with a diameter of 56 mm and a thickness of 2 mm was mounted on the cathode, meeting the characteristics of the DBD plasma generation technique.

The acrylic tube was 110 mm in height, 80 mm in outer diameter, and 74 mm in inner diameter. It penetrated 5 mm in each Teflon flange, allowing a maximum distance of 100 mm between the electrodes. There was a rack that allowed 0.1 mm steps to adjust the distance between the electrodes [15]. 

The power source used in the reactor provided a voltage of 20 kV, a frequency between 200 and 1000 Hz, and a pulse width of the order of 200 μs.

### 2.3. Treatment

The treatments consisted of the plasma generated between the electrodes using argon and atmospheric pressure. The DBD was formed from the parameters shown in Table 1. Frequency monitoring was performed through a digital oscilloscope (Keysight DSO01072B).

### 2.4. Characterizations

The characterization techniques used in this work were: wettability test to determine the surface tension, infrared spectroscopy with Fourier transform by attenuated total reflectance (ATR-FTIR, Bruker FT-IR Vertex 70, Durham, UK, X-ray excited photoelectrons spectroscopy (XPS, ThermoFísher K-Alpha, Waltham, MA, USA), atomic force microscopy (AFM, Shimadzu SPM-9700, Kyoto, Japan), and the triple probe method of Langmuir (LP3–LabPlasma UFRN, Natal, Brasil), to quantify the energy generated by the plasma during the treatments.

The wettability test was carried out by determining the contact angle between the drop (20 µL) and the sample surface before and after treatments using the sessile drop method [49,50]. For the calculation of surface tensions, the Fowkes method [51] was used from the contact angle of three liquids: water, glycerol, and dimethylformamide. The test was done before, shortly after, and 2 and 7 days after treatment to study aging over time for each treatment [52,53].

The ATR-FTIR equipment from Bruker model FTIR VERTEX 70 was used to characterize the chemical groups of the sample surface [54,55] spectra in the range of 400 to 4000 cm^−1^.

The XPS returned the spectrum of 1 up to 1360 eV, determining the power band of the elements and their percentage. It was also possible to observe the chemical bonds formed and quantify them, using a Thermo Fisher K-alpha X-ray photoelectron spectrometer equipment.

The AFM was used to characterize the surface morphology and the roughness, using a Shimadzu SPM-9700.

The Langmuir Triple probe is a disseminated technique for the diagnosis of plasma, which can provide the electronic temperature, i.e., the average energy of the electrons [56,57]. Then, it is possible to correlate the average energy of the electrons with the energy of the chemical bond, estimating the functional groups generated superficially during the plasma treatments [58,59,60,61].

## 3. Results

The plasma electron temperature, obtained from the triple probe of the Langmuir technique, is presented in Figure 2. The energy was calculated during the treatments, and the procedures are described elsewhere [61]. The behavior observed came from the plasma in the filamentary regime, featuring high-energy electrons during the peaks. The average energy collected during the treatments was 16.31 (in argon plasma) and 15.04 eV (in oxygen plasma), as presented in Table 2. Moreover, this behavior was predicted by the speed (energy) distribution of Boltzmann [62].

The energy values measured by this technique proved to be sufficient in altering the chemical bonds present in the HDPE chain, as shown in Table 2 [63]. It is thus possible to promote the breaking of surface bonds, as well as the incorporation of species, resulting in new functional groups.

The wettability test determined the angles measured with distilled water, glycerol, and dimethylformamide, before, right after, and two and seven days after treatment. Figure 3 lists the contact angles of water measured for each condition. Table 3 shows other angles and liquids surface tensions.

A decrease in the value of the angles was observed after all treatments, indicating a surface modification, as reported in the literature [37,59,64]. Then, as illustrated in Figure 4, to the decrease of the contact angle, the surface tension solid–gas (γ_SG_) increased [65]. This figure shows the drop behavior in contact with the surface before and after the plasma treatment. With time, the values approach the angle measured before being treated, emphasizing the aging process. This result confirms what has already been reported for other studies that modified polymeric materials by plasma at low pressure or atmospheric pressure [3,9,16,42,59,66,67]. Because the bonds generated by plasma treatment are formed only on the HDPE surface, the mobility of the polymeric chain and reaction of these polar groups with the environment humidity can promote the breakage of this bond or the stabilization of the same [3,9,16,42,66].

Based on the contact angles, the interfacial stresses were calculated by the Fowkes method. Thus, in Figure 5, the polar, dispersive, and total surface tension values of each treatment are presented. The surface tension values point to broken bonds such as C–C and emphasize the permanent character of the changes that occurred in the dispersive coordinates because, in seven days, there were no changes in the values of this surface tension coordinate. However, The increase in polar coordinates suggests the formation of oxygen bonds, C–O or C=O, as well their reduction with time suggests that these groups were, in some way, stabilized by reactions with the medium, as reported in the literature [3,9,16].

To investigate the possible chemical changes in the surface structure of HDPE, based on electronic temperature and wettability, ATR-FTIR analysis was performed. Figure 6 shows the results of ATR-FTIR for the samples treated with oxygen, argon, and those untreated. It is possible to observe the well-defined bands at 2915 (C–H asymmetrical) and 2847 cm^−1^ (C–H symmetrical). The band at 1462 cm^−1^ is associated with –CH_2_–vibrations, and the band at 720 cm^−1^ is linked to the C–C vibration [23,31]. The calculation of relative percentages of the bands was performed to determine which bond was broken and formed in the treated samples. Some bands that appeared in the treated samples, and most prominent in the treatment with oxygen, as seen at 1019 cm^−1^, are a characteristic of C–C and C–O vibrations. The peaks at 1742 and 1645 cm^−1^ are caused by C=O and C=C vibrations, respectively [54,68].

Figure 7 shows the calculation of the relative area percentages for the untreated sample, treated with argon, oxygen, and aged (two days) for each treatment. It is possible to visualize that the characteristic bands of HDPE (symmetrical and asymmetric C–H, CH_2_, and C–C) have a reduction in their area for both argon and oxygen, which results in the formation of the other connections indicated in the FTIR, which implies increasing wettability. It is also possible to see that two days after the treatment, there was an increase in the band associated with the C–H bond, indicating that, as predicted by the surface tension analysis, there is a reaction of the functional groups formed by the plasma treatment on the HDPE surface with the environment, this being the main factor for the reduction of wettability.

The relative area calculation of the bands reported after the treatment can be seen in Figure 8. The connections involving oxygen (C–O–C, C–O, C=O, OH) influence the polar coordinate. With aging time, there is a loss of the bonds formed, directly impacting the reduction of wettability [27,59].

Based on the changed bonds, and the percentage on the plasma energy (16.3 eV) calculated with the Langmuir probe and the HDPE chain. Figure 9 shows how chemical changes occur during plasma treatment.

XPS was used to quantify the elements and chemical bonds for specimens treated with argon and untreated. The introduction of argon decreases the partial pressure of atmospheric gas. However, the treatments were done at atmospheric pressure, and this means that it still had atmospheric air along with an ozone formation, which is the agent responsible for the incorporation of oxygen on the HDPE surface. Figure 10 shows the spectrum before and after treatments. Here, it is possible to see the introduction of the band related to O 1s. Figure 10B illustrates the appearance of the O1s characteristic band, directing the investigation to the formation of some bindings such as C–O and C=O. It is possible to affirm the increase in the percentages of the peaks related to these bonds involving carbon and oxygen. The C1s spectrum shows a peak at 285 eV, which is characteristic of C–C/C–H bonds. In the treated samples, a peak at 288 eV can be observed, which is attributed to O–C–O and C=O bindings. Another peak founded out in the analysis was O1s. The peaks highlighted were around 332.5 and 334.6 eV, which are attributed to the O–C–O and O–C=O binding, respectively [23,58,69,70,71].

Figure 11 shows the relative percentages averages of HDPE bindings to compare the argon treated sample with untreated sample and, as like FTIR, the emergence of oxygen groups and a reduction in the percentage of carbon can be seen, signaling the break of connections at peak~285 eV and the formation of connections such as C–O and C=O when peaks 288, 332.5, and 334.6 eV increase their relative percentage.

In the XPS, the energy bands in the binding involving oxygen are much more evident than in the FTIR, showing that the modification is minimal in the material structure, being only in the surface threshold, as illustrated in Figure 12. With this, it promotes the instability of these connections guiding the cause of the treatment aging, evidenced by the wettability test to the lack of surface stability.

The atomic strength microscopy characterized the surface morphology. The details related to the increase and diminish surface roughness were compared with profile rugosity parameters. Figure 13 shows the AFM in a 5 × 5 µm^2^ area for untreated samples (Figure 13A), argon treated (Figure 13B), and oxygen treated (Figure 13C). The plasma treatment increased the roughness, which was more evident for argon. The physical and chemical modifications change the surface wettability. Table 4 shows the mean roughness to quantify the morphological improvements. 

Since the chemical modifications have aged over time, influencing the polar coordinate more clearly, due to the formation of functional groups with oxygen, the physical modification tends to be maintained, with the aging of morphological modifications not evident in polymers [1,33]. This change in surface roughness is due to the etching action promoted by the species present in the plasma, as reported in the literature [67,72,73,74,75].

The etching is due, in its majority, to species with high kinetic energy, confirmed by the electronic temperature, which collides with the HDPE surface generating a material erosion. It is observed that the most significant roughness was for the argon plasma, revealing even more the inert character of this mechanism. Additionally, it is noteworthy that because it is a DBD plasma in the filamentary regime, the concentration of plasma in some areas of the sample contributes to the potentiation of this effect. With this, the dispersive coordinate is related to the increase in roughness.

## 4. Conclusions

The plasma generated in these treatment conditions has the necessary energy for modifying the polymeric material surface. It generated an electronic temperature of approximately 16 eV, proving to be sufficient to promote the break of C–C and C–H, the formation of C–O, O–C–O, O–C=O, and C=O bindings. These formed bonds are responsible for increasing the wettability of the material by increasing the polar coordinate. However, they were only formed on the surface and tended to suffer from aging. The material tends to return to its original wettability. Although plasma treatment promotes an increase in HDPE surface roughness, which is related to the increase in surface tension dispersive coordinate, this factor is not sufficient to maintain the hydrophobicity of HDPE after plasma treatment.

## Figures and Tables

**Figure 1 polymers-12-02422-f001:**
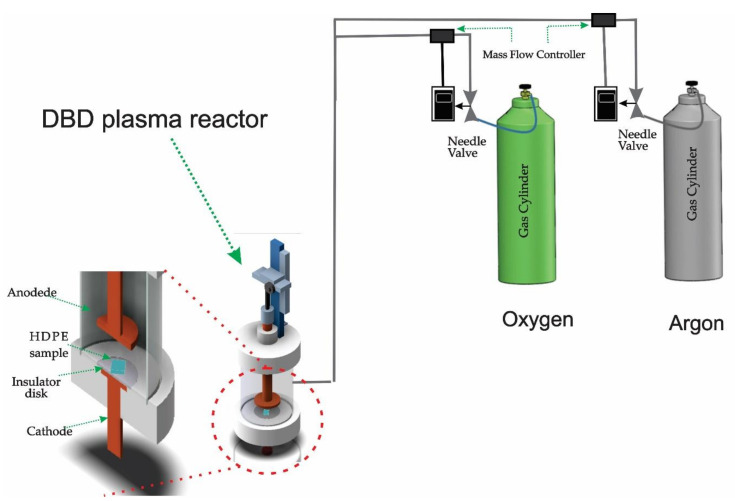
Dielectric barrier discharge (DBD) plasma reactor.

**Figure 2 polymers-12-02422-f002:**
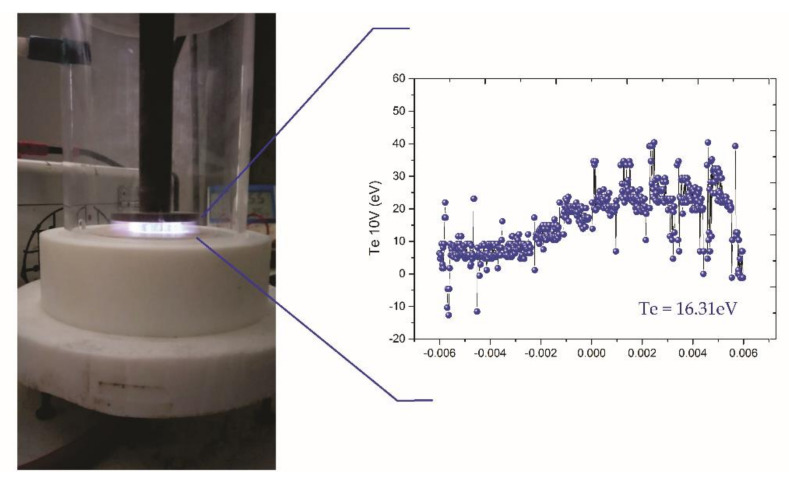
High-density polyethylene (HDPE) during the treatment and the behavior of the electronic temperature of the plasma.

**Figure 3 polymers-12-02422-f003:**
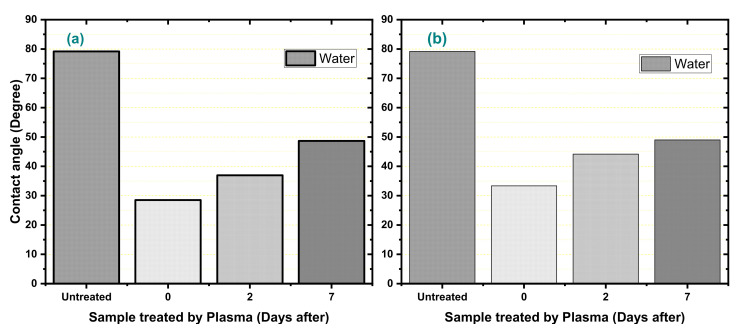
Values of the angles of water obtained before and after the treatment in (**a**) argon and (**b**) oxygen.

**Figure 4 polymers-12-02422-f004:**
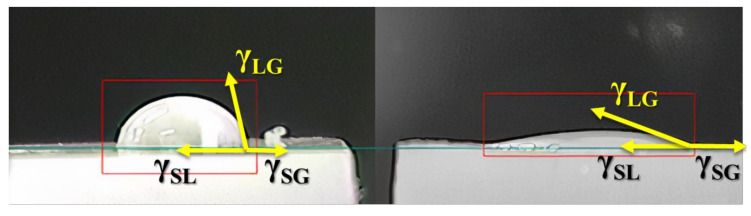
The behavior of the drop in contact with the surface of material before and after treatment.

**Figure 5 polymers-12-02422-f005:**
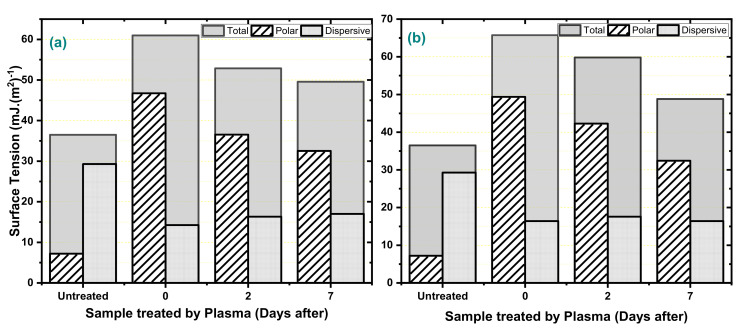
HDPE superficial tensions over time in (**a**) argon and (**b**) oxygen.

**Figure 6 polymers-12-02422-f006:**
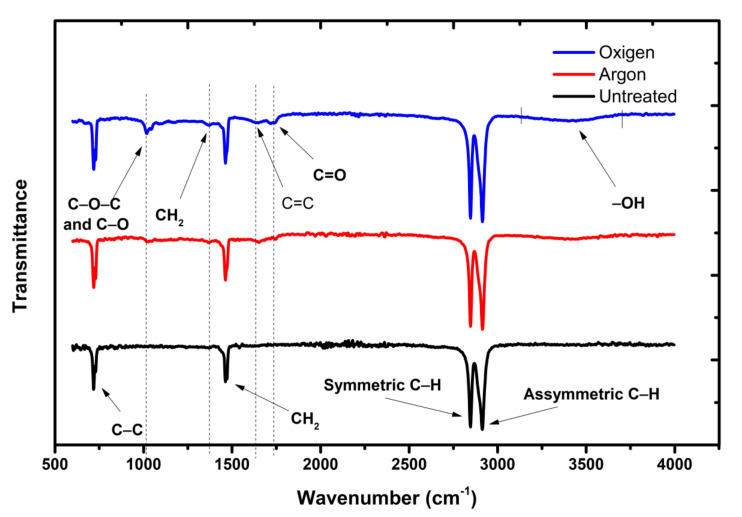
FTIR of the treated samples and the standard sample.

**Figure 7 polymers-12-02422-f007:**
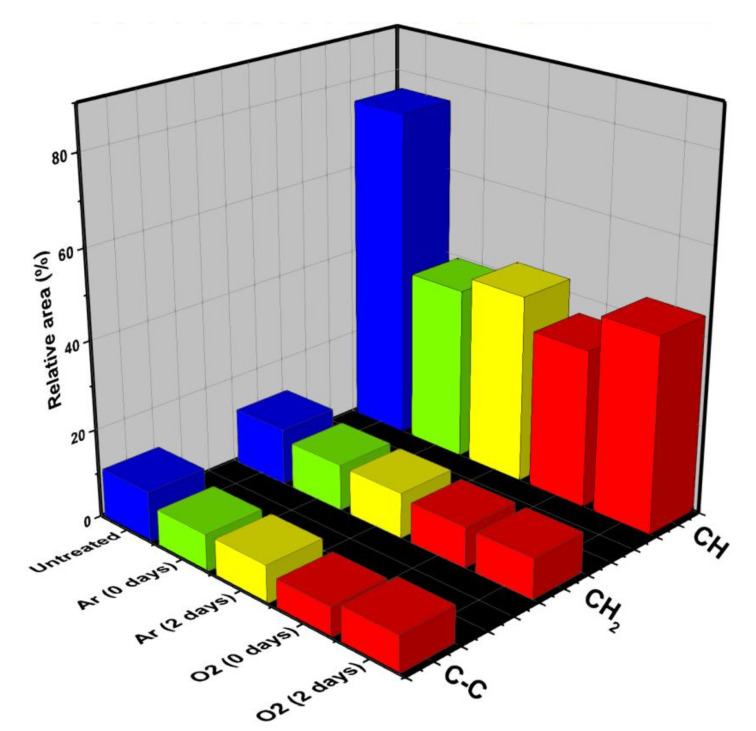
Relative percentages of regions of the main peaks.

**Figure 8 polymers-12-02422-f008:**
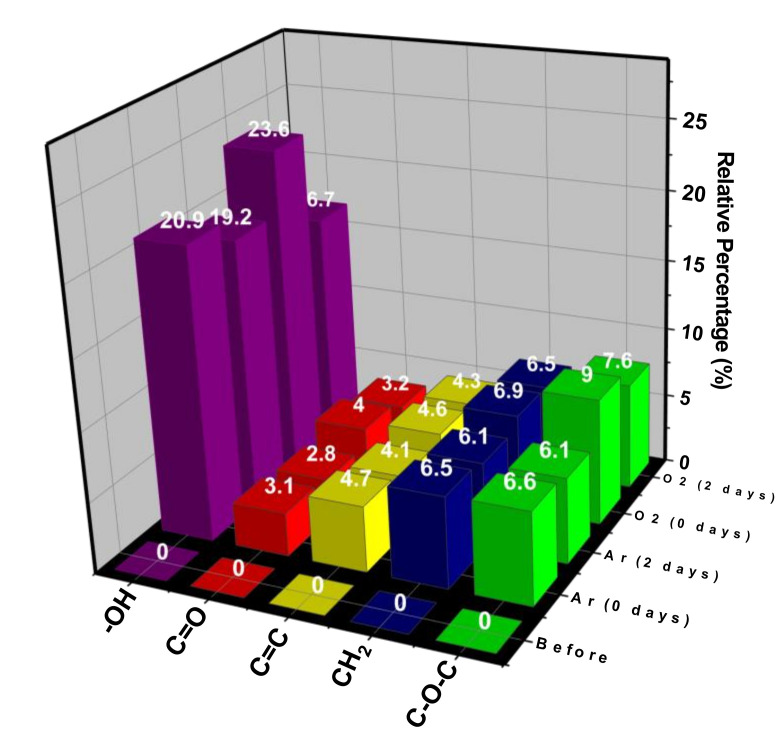
Relative percentages of regions of the other peaks.

**Figure 9 polymers-12-02422-f009:**
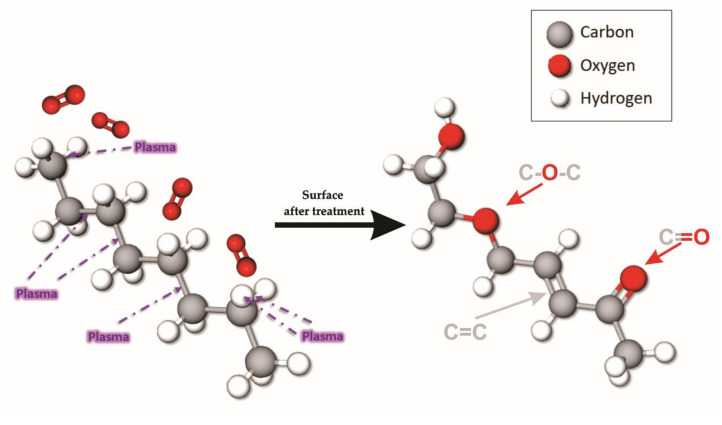
Chemical bonds altered during plasma treatment.

**Figure 10 polymers-12-02422-f010:**
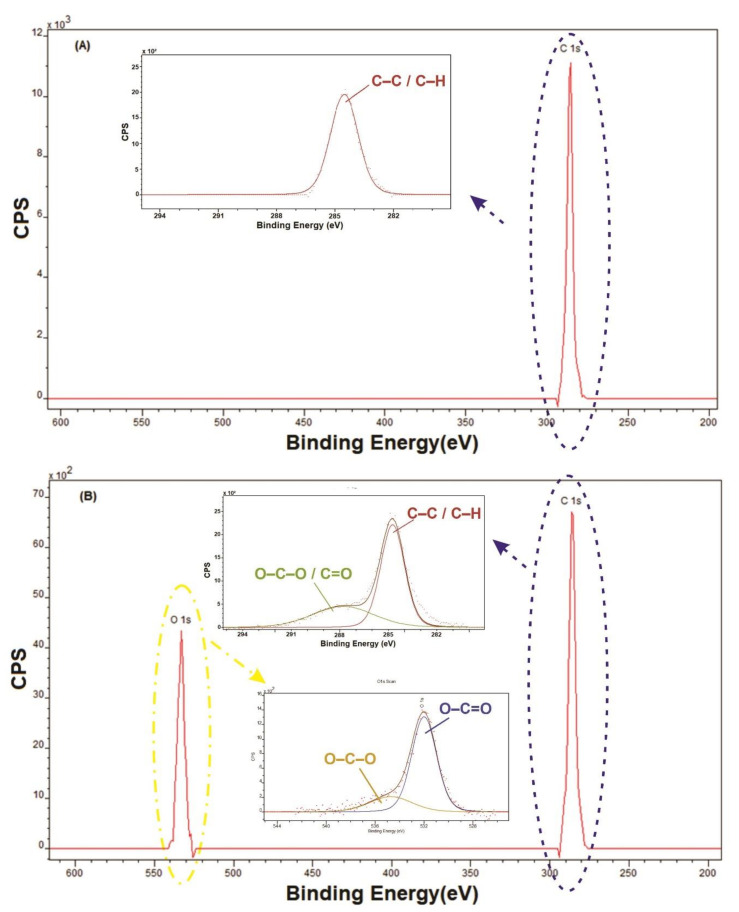
XPS analysis (**A**) before and (**B**) after treatment of argon plasma treatment.

**Figure 11 polymers-12-02422-f011:**
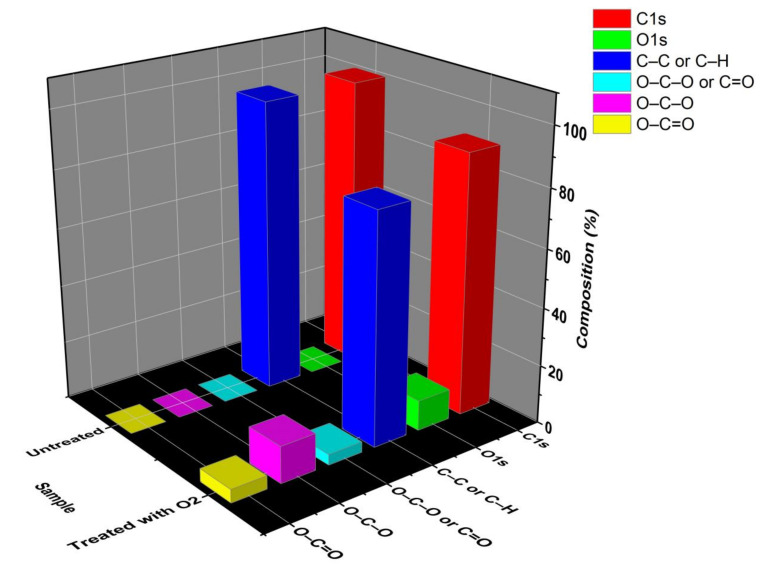
Average percentages of the binding obtained by deconvolutions.

**Figure 12 polymers-12-02422-f012:**
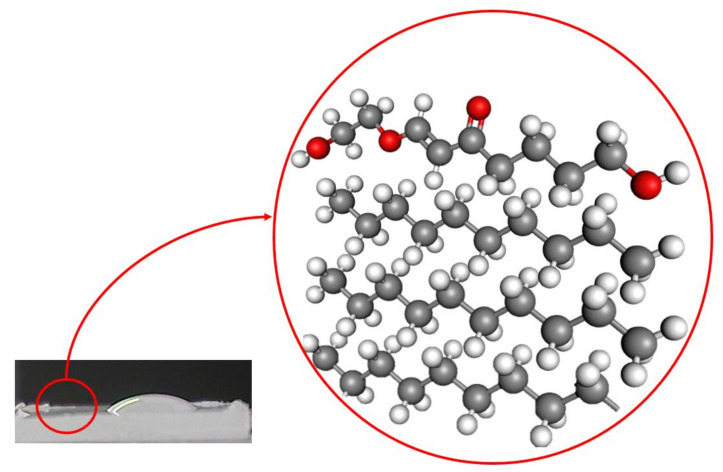
HDPE surface modification by plasma DBD treatment.

**Figure 13 polymers-12-02422-f013:**
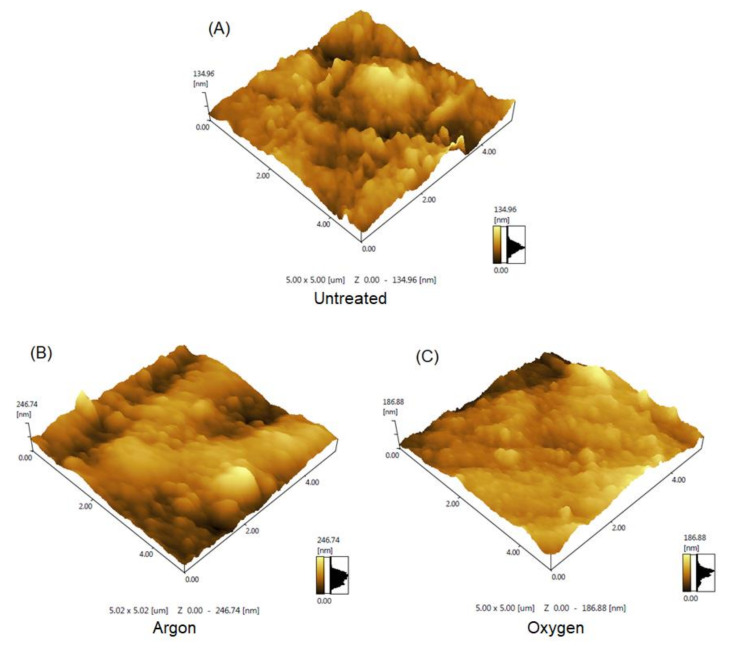
AFM of HDPE samples (**A**) untreated, (**B**) treated with an argon plasma, (**C**) treated with oxygen plasma.

**Table 1 polymers-12-02422-t001:** Parameters used for the treatments.

Applied Voltage (kV)	20
Frequency (Hz)	820
Distance between the Electrodes (mm)	1.7
Time treatment (s)	600

**Table 2 polymers-12-02422-t002:** Calculated plasma energy and energy of chemical bonds.

Mean Energy Calculated During the Plasma Treatments		Bonds that May Be Influenced by Plasma Treatment	Connection Energies (eV)
Argon	Oxygen		
16.31 (eV)	15.04 (eV)	C–H	4.29
		C–C	3.60
		C–O	3.64
		C=O	7.37
		C=C	6.33

**Table 3 polymers-12-02422-t003:** Values total, polar (γ_l_^p^) and dispersive (γ_l_^p^) coordinates of superficial tensions, and the values of contact angle for the liquids.

Liquid	Surface Tensions (mJ/m^2^)			Contact Angle (Degrees)						
	γ_l_	γ_l_^p^	γ_l_^d^	Untreated	Argon			Oxygen		
					0 Days	2 Days	7 Days	0 Days	2 Days	7 Days
Water	72.8	51.0	21.8	79.17	28.52	36.94	48.66	33.34	44.12	49.02
Dimethyl-formamide	37.3	4.9	32.4	31.72	0	0	0	0	0	15.49
Glycerol	63.4	26.2	37.2	53.17	9.91	24.75	52.75	37.63	45.34	46.63

**Table 4 polymers-12-02422-t004:** Average roughness obtained from treated and untreated samples.

	Untreated	Argon	Oxygen
Ra (nm)	14.084	32.314	16.659

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
