# Peer review of "Study of High-Density Polyethylene (HDPE) Kinetics Modification Treated by Dielectric Barrier Discharge (DBD) Plasma"

_polymers, 2020, doi:10.3390/polym12102422_

Round 1

Reviewer 1 Report

As attached.

Author Response

We are very grateful to the honorable reviewer for highly useful comments and suggestions. We agree with the honorable reviewer, and we try to improve the discussion of the results, addressing the points suggested by the honorable reviewer. Please note the texts added on pages 05 (lines 145 - 150), page 06 (lines 158 - 161), and page 13 (lines 244 - 251).

Reviewer 2 Report

The manuscript by these authors is an interesting piece deal with spectroscopic characterization aiming to investigate the physico-chemical changes of HDPE surface. The experimental aspect of the work is really interesting, the authors carried out a deep investigation well discussed in the text and the results are worthy of publication. Anyway, the text needs an important revision in order to remove many mistakes and correct some typos (both highlighted in the attached .pdf). I would also suggest the addition of some references, namely: in the introduction in order to strengthen some aspects and in the experimental to support the method used for calculation (both are highlighted in the attached .pdf). I think that after this minor revision the manuscript can be published on polymers.

Author Response

We are very grateful to the honorable reviewer for highly useful comments and suggestions. We agree with the honorable reviewer, and we try to improve the text by summarizing by expressing the importance of our results. We hope to have met the esteemed reviewer.

We hope to have answered any suggestions. We would like to thank the didactic, professional and efficient way of addressing the corrections that improve the quality of work.

The honorable reviewer can view as highlighted changes in the submitted manuscript.

Reviewer 3 Report

The paper entitled "Study of HDPE kinetics modification treated by DBD plasma" by João Freire de Medeiros Neto and co. describes the modifcation of the high-density polyethylene (HDPE) surface by DBD plasma treatment in different gases.

The subject of the paper has been widely debated in the literature and the present work doesn't present any novelty. 

The authors used only one polimeric sample, treated in argon or oxygen plasma and in the section "Materials and methods" it is not clear the provider of the polymer.   It is not a new obtained polymer and it is not a new method of obtaining surface modification, so the conclusions are not original.

It is not clear what is the main scientific meaning of the paper, what was the scope of the work. The authors concluded the "break of C-C and C-H, and formation of C-O, O-C-O, O- C=O and C=O bindings" and present this fact as as a novelty, as well as the "formed bonds are responsible for increasing the wettability of the material by increasing the polar coordinate". 

Author Response

We are very grateful to the honorable reviewer for highly useful comments and suggestions. We agree with the honorable reviewer, and we try to improve the text by summarizing by expressing the importance of our results. We hope to have met the esteemed reviewer.

We agree that the subject of the article has been widely debated in the literature, but this present work presents (in our opinion), some relevant news. In the last paragraph of the introduction (line 32), this exhibition the main novelty of the present work is the use of the dielectric barrier discharge (DBD) technique in High-Density Polyethylene (HDPE) which, although already worked in other research centers. Our approach was intended to go further in depth on the surface modification kinetics of this polymer when subjected to a plasma atmosphere.

 Another innovative approach of this work is the use of electronic temperature data by Triple Langmuir Probe, a technique already dominated by our research group (see reference 57), results that are beginning to be discussed in line 118 and provide a better basis for why it was possible to modify the surface of the material in question based on the energy present in the plasma.

 With this, I believe that the main objective of our work was well described by the title of it, that is, the better understanding of how to increase wettability not in any polymer but in HDPE using with the explanation in addition to what has already been said as well as the connections formed and consequently the change in the polar coordinate.

Round 2

Reviewer 3 Report

The subject of the joutrnal is the polymer study, but the authors claim the originality of electron energy/temperature measurements. A journal concerning plasma techinques will be more suitable. Only onel sample of polymer was studied.

Moreover, usually the influence of heavy particle (ions) from plasma is studied for explain the polimeric surface modification. Such particles can produce degradation or can be involved in polymer chain breaking due to their energy gaigned in plasma and high mass. This article did not take in consideration this aspect.

Some studies  stipulates the presence of two groups of electrons with different energies demosntrated by Langmuir peobe measurements. Did you take n consideration this aspect? Why you assume the Boltzmann distribution in the studied plasma configuration?

Author Response

We are very grateful to the honorable reviewer for highly useful comments and suggestions. We agree with the honorable reviewer, and we try to improve the text by summarizing by expressing the importance of our results. We hope to have met the esteemed reviewer. Dear reviewer, in this article, the study of electronic temperature by Langmuir triple probe was not carried out. Only the knowledge of it and the data provided to more consistently guide the effects on the modification of polymeric materials. And yes, only HDPE was used in this work; however, several samples were processed by DBD plasma, at least five samples for each analysis (wettability, XPS, AFM). we added this information on page 2 (lines 76-77)

Our work on electronic temperature has already been published in a magazine specialized in this subject with the title “Triple Langmuir probe, optical emission spectroscopy and Lissajous figures for diagnosis of plasma produced by dielectric barrier discharge of parallel plates in atmospheric pressure,” DOI: 10.3233 / JAE-190044, it contains in detail why the use of Boltzmann distribution.

We agree with the honorable reviewer that an article dealing with plasma physics is more suitable for the scope of another journal. However, Polymers Magazine is publishing in a special edition "Plasma Processes for Polymers," so we understand that the subject addressed in this work is relevant. Therefore we used some elements of plasma physics in a way that only complemented our discussion. As explained, we deal with plasma physics in article DOI: 10.3233 / JAE-190044.

Regarding the influence of heavy particles present in the plasma and its relationship with the changes promoted in the polymeric materials, the two reviewers questioned us about the theme. They were answered with the addition of the texts on pages 05 (lines 145 - 150), page 06 (lines 158 - 161), and page 13 (lines 244 - 251). After this change, they accepted the explanation, recognizing the centric merit of the paper and by the discussions presented.

We make ourselves available for any further clarification, we appreciate the attention given to work, and we recognize that the suggestions made by the honorable reviewer enhance our work.

Respectfully…

Round 3

Reviewer 3 Report

I carrefully read the paper again. I am still remaining to some questions:

1) In the first response you said:

"Another innovative approach of this work is the use of electronic temperature data by Triple Langmuir Probe, a technique already dominated by our research group (see reference 57),...."

But now you said 

"Dear reviewer, in this article, the study of electronic temperature by Langmuir triple probe was not carried out."

So, it is innovative but not described in this article. 

2) Page 2 line 76 " In this work were treated five samples to each treatment condicion and the results showed is the average obtained in the analysis. "

I do not understand the methodology of the research you use. 

For a single treatment condition, you obtained five different results and you present the average. So, the treatment it is not reproductible if for the same condition you have five different results. 

Different samples- I understand different conditions (different polymeric sample or different plasma treatment input conditions). 

3) Regarding the heavy particles the response was:

"This result confirms what has already been reported for other  studies that modified polymeric materials by plasma at low pressure or atmospheric pressure [3,9,16,42,59,66,67]. Because the bonds generated by plasma treatment are formed only on the HDPE surface, the mobility of the polymeric chain and reaction of these polar groups with the environment humidity can promote the breakage of this bond or the stabilization of the same [3,9,16,42,66]"

The papers you cited used other polymers or other plasma configuration, how can you be sure that their results are suitable for your work?

Author Response

We hope to answer and clarify the points addressed by the honorable reviewer in the text below

1) In the first response you said:

"Another innovative approach of this work is the use of electronic temperature data by Triple Langmuir Probe, a technique already dominated by our research group (see reference 57),...."

But now you said

"Dear reviewer, in this article, the study of electronic temperature by Langmuir triple probe was not carried out."

So, it is innovative but not described in this article.

ANSWER: We are very grateful to the honorable reviewer for highly useful comments and suggestions. We agree with the honorable reviewer. The study of the Langmuir triple probe was carried out in the article. We expressed ourselves poorly in the response. The use of this technique made possible the debate about the energy of species present in the plasma responsible for the modification of HDPE. However, the debate over the foundation of the Langmuir triple probe technique is a matter of plasma physics, which, as suggested by the honorable reviewer, could be, and has been, discussed in another journal.

2) Page 2 line 76 " In this work were treated five samples to each treatment condicion and the results showed is the average obtained in the analysis. "

I do not understand the methodology of the research you use.

For a single treatment condition, you obtained five different results and you present the average. So, the treatment it is not reproductible if for the same condition you have five different results.

Different samples- I understand different conditions (different polymeric sample or different plasma treatment input conditions).

ANSWER: We are very grateful to the honorable reviewer for highly useful comments and suggestions. To guarantee the reproducibility of the technique, each treatment was repeated five times, resulting in 5 samples for each gas (the treatment condition changes, as it changes the gas in the plasma atmosphere). However, note that several analyzes were performed, FITR, XPS, contact angle, AFM. The results of each analysis were very close, which guarantees the reproducibility of the treatment. Remember that this is a methodology generally used in experimental research.

3) Regarding the heavy particles the response was:

"This result confirms what has already been reported for other  studies that modified polymeric materials by plasma at low pressure or atmospheric pressure [3,9,16,42,59,66,67]. Because the bonds generated by plasma treatment are formed only on the HDPE surface, the mobility of the polymeric chain and reaction of these polar groups with the environment humidity can promote the breakage of this bond or the stabilization of the same [3,9,16,42,66]"

The papers you cited used other polymers or other plasma configuration, how can you be sure that their results are suitable for your work?

ANSWER: Honorable reviewer, as mentioned, the articles address the use of PLASMA TECHNIQUE in the modification of other polymers and with other plasma configurations. However, note that the kinetics of the species' behavior is the same, as they are physical-chemical reactions promoted by species present in the plasma. This implies that regardless of the polymer and the plasma configuration if there were species with adequate energy, the changes would be the same, as they are inherent to the processing of polymers by plasma.